:PLOS | ONE

# Burden and risk factors of cutaneous leishmaniasis in a peri-urban settlement in Kenya, 2016

Isaac Ngere[1,2☯]*, Waqo Gufu Boru[2,3☯], Abdikadir Isack[2,3‡], Joshua Muiruri[2,3‡], Mark Obonyo[2,3‡], Sultani Matendechero[3‡], Zeinab Gura[2,3☯]

**1** Global Health Program-Kenya, Washington State University, Nairobi, Kenya, **2** Field Epidemiology and Laboratory Training Program, Nairobi, Kenya, **3** Ministry of Health, Nairobi, Republic of Kenya

☯ These authors contributed equally to this work.
‡ These authors also contributed equally to this work
* isaac.ngere@wsu.edu

**Data Availability Statement:** All relevant data are within the manuscript and its Supporting Information files.

## Abstract

### Background

Cutaneous leishmaniasis is a neglected disease known to cause significant morbidity among the poor. We investigated a suspected outbreak to determine the magnitude of cases, characterize the cases and identify risk factors of cutaneous leishmaniasis in Gilgil, a peri-urban settlement in Central Kenya.

### Methods

Hospital records for the period 2010–2016 were reviewed and additional cases were identified through active case search. Clinical diagnosis of cutaneous leishmaniasis was made based on presence of ulcerative, nodular or papular skin lesion. The study enrolled 58 cases matched by age and neighbourhood to 116 controls in a case control study. Data was collected using structured questionnaires and simple proportions, means and medians were computed, and logistic regression models were constructed for analysis of individual, indoor and outdoor risk factors.

### Results

Of the 255 suspected cases of cutaneous leishmaniasis identified, females constituted 56% (142/255) and the median age was 7 years (IQR 7–21). Cases occurred in clusters and up to 43% of cases originated from Gitare (73/255) and Kambi-Turkana (36/255) villages. A continuous transmission pattern was depicted throughout the period under review. Individual risk factors included staying outside the residence in the evening after sunset (OR 4.1, CI 1.2–16.2) and visiting forests (OR 4.56, CI 2.04–10.22). Sharing residence with a case (OR 14.4, CI 3.8–79.3), residing in a thatched house (OR 7.9, CI 1.9–45.7) and cracked walls (OR 2.3, CI 1.0–4.9) were identified among indoor factors while sighting rock hyraxes near residence (OR 5.3, CI 2.2–12.7), residing near a forest (OR 7.8, CI 2.8–26.4) and

**Funding:** This field investigation was funded by the Kenya Ministry of Health (MoH), Field epidemiology and laboratory training program (KFELTP). The KFELTP is a competency-based program with mandate to build epidemiological capacity undertaken jointly by the of Ministries of Health and that of Agriculture, Livestock and Fisheries (MoALF) by offering scholarships to employees within the two government ministries to pursue a two-year masters degree program in field epidemiology. Among the competencies needed to be achieved by the trainees is participating in an outbreak investigations in-country or abroad. The KFELTP mentors who are co-authors in this paper, were assigned to mentor the corresponding author at that time of the investigation and they assisted in designing the investigations and conducting preliminary analysis and generation of outbreak report. However the KFELTP had no role in data collection, decision to publish or preparation of the manuscript.

**Competing interests:** The authors have declared that no competing interests exist.

having a close neighbour with cutaneous leishmaniasis (OR 6.8, CI 2.8–16.0) were identified among outdoor factors.

## Conclusions

We identify a large burden of cutaneous leishmaniasis in Gilgil with evidence of individual, indoor and outdoor factors of disease spread. The role of environmental factors and rodents in disease transmission should be investigated further

## Introduction

Leishmaniasis is a disease caused by a protozoan, *Leishmania* and is transmitted to humans and other mammals by the bite of a female phlebotomite sand-fly (*Phlebotomus species*). Three forms of the disease affect humans; cutaneous, muco-cutaneous and visceral forms (*Kala-azar*). The disease is considered a neglected tropical disease mainly affecting the rural poor. Cutaneous leishmaniasis (CL) commonly occurs in clusters among destabilized or migrant populations in low socio-economic settings with the current trend in distribution of new infections indicating a progressive spread of the disease to previously non-endemic areas [1–5].

Worldwide, over 350 million people are estimated to be at risk of CL and up to 1.5 million new infections are reported annually [6]. Owing to challenges in surveillance and reporting, the burden of CL is grossly underestimated [4]. Though disfiguring and debilitating in the affected people, the disease is rarely fatal, hence little attention has been given to prevention and control measures by health authorities [4,5]. Despite that, proven control strategies including vector eradication and early treatment of insect bites in endemic areas have been shown to be successful. Insect vector control activities such as indoor insecticide residual sprays, insecticide impregnated barriers (bed nets, curtains, clothes, carpets), environmental spraying, and control of reservoir hosts (rodents) are effective but expensive when rolled out on a large scale [4]. Therefore, targeted control programs guided by an understanding of local drivers of the disease including lifestyle and environmental factors would provide significant cost savings and value for money in addition to achieving disease control [5,7].

Recurrent outbreaks of CL in Kenya, Ethiopia and South Sudan have been reported in the past and are often associated with high morbidity. Kenya is classified by WHO as endemic for CL. However, there is relative scarcity of published data on the extent, burden and risk factors of CL [8]. In other parts of the world, urbanization and expansion of farming and other human activities into forests is often associated with disease outbreaks [5]. In Kenya, areas around the Rift Valley escarpments and major mountains are known natural habitats for sand-flies [9–11]. More than 48 species of sand-flies, including the *Phlebotomus species* that are vectors for CL (*P. duboscqi, P. guggisbergi, P. pedifer* and *P. acleatus*), have been identified in various habitats [12]. Recently, the areas around the Rift Valley in Kenya have been experiencing rapid population growth and increased environmental pressure resulting from in-migration and increased human activities in forests [13,14].

In early 2016, the health ministry of Kenya received notification about increase in cases of a skin disease suspected to be CL in Nakuru county, in south-eastern Rift Valley. We report the findings of a records review and a follow-up case control study conducted to determine the magnitude of the disease, characterize the cases and identify factors associated with the spread of the disease.

## Methods

### Study site

The study was conducted in Gilgil sub-county, a rapidly growing peri-urban settlement located in south-eastern part of the Great Rift Valley in Kenya, between 20[th] January and 3[rd] February 2016 (**Fig 1**). The terrain of Gilgil sub-county is generally mountainous to the north with plenty of rocky escarpments. The southern end of the sub-county is mainly composed of undulating plains, flat grazing lands and solidified lava that form large crevices and rocky caves infested with wild mammals and rodents [9,11]. The area is sparsely populated (population density of <200 persons per square Km) and typical scattered housing characterizes the settlement pattern this area [15]. Of late, the area has experienced an influx of new settlements since it is regarded as a high potential area yet a cheaper alternative to the urban life in neighbouring Nairobi city or Nakuru town [14]. The sub-county is traversed by a busy highway and is a preferred destination for potential peri-urban home-owners due to its ease of access from the surrounding urban centres. Previous studies in this area have identified the insect vector (*Sand-fly*) and the agent (*Leishmania donovani*, *Leishmania infantum* and *Leishmania chagasi*) to be prevalent in this area[9–11].

### Study design

Hospital records were reviewed by trained staff and additional cases were identified through door-to-door case search. Cases were then enrolled in a population-based case control study.

### Review of records

A standard data abstraction tool was used to review records covering nine health facilities in Murindati and Mbaruk/Eburru wards to develop an outbreak line-list. The facilities included Afya medical clinic, Camp Brethren medical clinic, Eburru dispensary, Mbaruk dispensary, Ol-Jorai health centre, Rhine Valley health center, Karunga dispensary, Langa-Langa dispensary and Anti-Stock Theft Unit dispensary. The total catchment population for these facilities is approximately 54,000 persons [18]. Entries made in outpatient, inpatient, laboratory, and specialist clinic registers between January 2010 and January 2016 were included in the records review.

A suspected case of CL was defined based on clinical diagnosis recorded in hospital records as 'skin ulcer', 'skin wound', 'plaque', 'dermatitis', 'skin infection' or 'cutaneous leishmaniasis'. The outbreak line-list was updated through addition of probable cases. A probable case was defined as a resident with a typical skin lesion (a skin ulcer with typical raised edges and depressed centre or a skin plaque-a circumscribed, nodular or palpable skin lesion) on physical examination by a medical officer in the study team during the study period. Due to delays in receiving sample collection and laboratory testing supplies, no laboratory confirmation for CL was done on the suspected or the probable cases. All entries that matched 'suspected', 'probable' or 'confirmed' case definitions were included in the outbreak line list. The line list also included other information such as name, sex, age, date seen at the facility, residence, signs and symptoms, diagnosis and treatment given. Patients whose clinical diagnosis and contact information (physical address or phone contact) were missing were excluded from the line list.

### Enrolment of cases and controls into a case control study

To determine the risk factors of CL infection in the study population, we conducted a follow-up case-control study. Cases and controls consisted of eligible residents found in the study area during the study period (20[th] January-3[rd] February 2016).

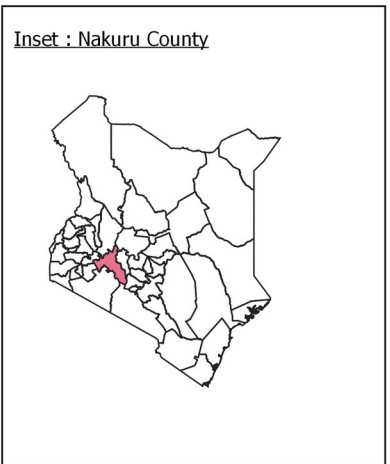

**Fig 1. Map of the investigation area, Gilgil, Kenya 2016.** This map was drawn on QGIS Version 2.18.15 using mapping resources from International Livestock Research Institute (ILRI) [16,17].

**Case and control recruitment.** We approached and enrolled 59 cases for the case control study: 41 cases who could be reached during the study period were selected from the outbreak line-list and were first ascertained to be probable CL cases by experienced medical officers in the study team on the basis of typical skin lesions (a skin ulcer with raised edges and depressed centre or a skin plaque, described as a circumscribed, nodular or papular skin lesion). A further 18 probable cases were identified from the community during active house-to-house survey upon examination by the medical officers (**Fig 2**). The investigation team comprising a field epidemiologist, 2 medical doctors, a laboratory scientist and 2 public health specialists, worked with community-based locators (community health volunteers and local chiefs) and the recruited cases to locate additional cases for inclusion in the case control study in a respondent-driven sampling process. In each village, the number of cases that were recruited in the case control study was allocated by probability proportional to size sampling approach based on the proportion of residents from that village with suspected CL from the outbreak line-list.

Each of the enrolled cases were matched to two community-based controls by age using the following criteria: Cases less than two years of age were matched to controls within two years, cases 2–4 years old were matched to controls within 3 years, cases 5–19 years to controls within 5 years, cases 20–59 year's old to controls within10 years, and cases more than 60 years old to controls within 20 years. One case was dropped in the final analysis owing to lack of suitable controls, bringing the total number of cases and controls included in the study to 174 (58 cases and 116 controls) (**Fig 2**).

Controls were selected from among residents of the same age group and living in the same or neighbouring village(s) as the case patients, and had no typical skin lesions (ulcer, plaque, wound or scar) upon inquiry and examination by medical doctors in the study team. To locate a possible control, a member of the investigation team would spin a bottle while standing afront the case's residence to determine the direction of movement. A random number between 2 and 5 was drawn to indicate the number of houses in the chosen direction to be passed before the team would attempt to recruit a control. This process was repeated until two eligible controls were recruited for each case. Five potential controls were dropped from the study on account of having old healed scars upon examination (**Fig 2**).

**Sample size.** The sample size was determined by the number of eligible cases present in the study area during the study period. A sample size of 58 patients and 116 controls (2 controls per case) was adopted to give the study at least 80% power at the 5% significance level and able to detect an odds ratio (OR) of 0.3 for an exposure present in 31% of controls [19,20]. The exposure chosen was use of mosquito nets.

**Data collection and analysis.** A structured questionnaire (study questionnaire in S1 Questionnaire) was developed and pretested in three randomly selected non-study households on the first day of field work. The questionnaire was administered to cases and controls through face to face interviews. Information on demographic profile, clinical presentation, risk factor profile and environmental exposures among cases and controls regarding CL were collected. Exposure data in both cases and controls was collected in relation to the year of onset of symptoms in the case patients. Environmental observations around the home were made by the study personnel and recorded in the questionnaires.

All data from the questionnaires were entered into a database and cleaned using Epi info version 7 (CDC, Atlanta GA, USA) and Microsoft Excel (2010). Simple proportions, means and medians were calculated for categorical data and continuous data respectively. To identify factors associated with the outbreak, chi-square tests were conducted for categorical data, and odds ratios and 95% confidence intervals (95% CI) computed. Risk factors were categorised into three groups in the analysis: factors relating to the individual, factors relating to the indoor environment and factors relating to the outdoor environment. Factors with a *P≤0.05* in the

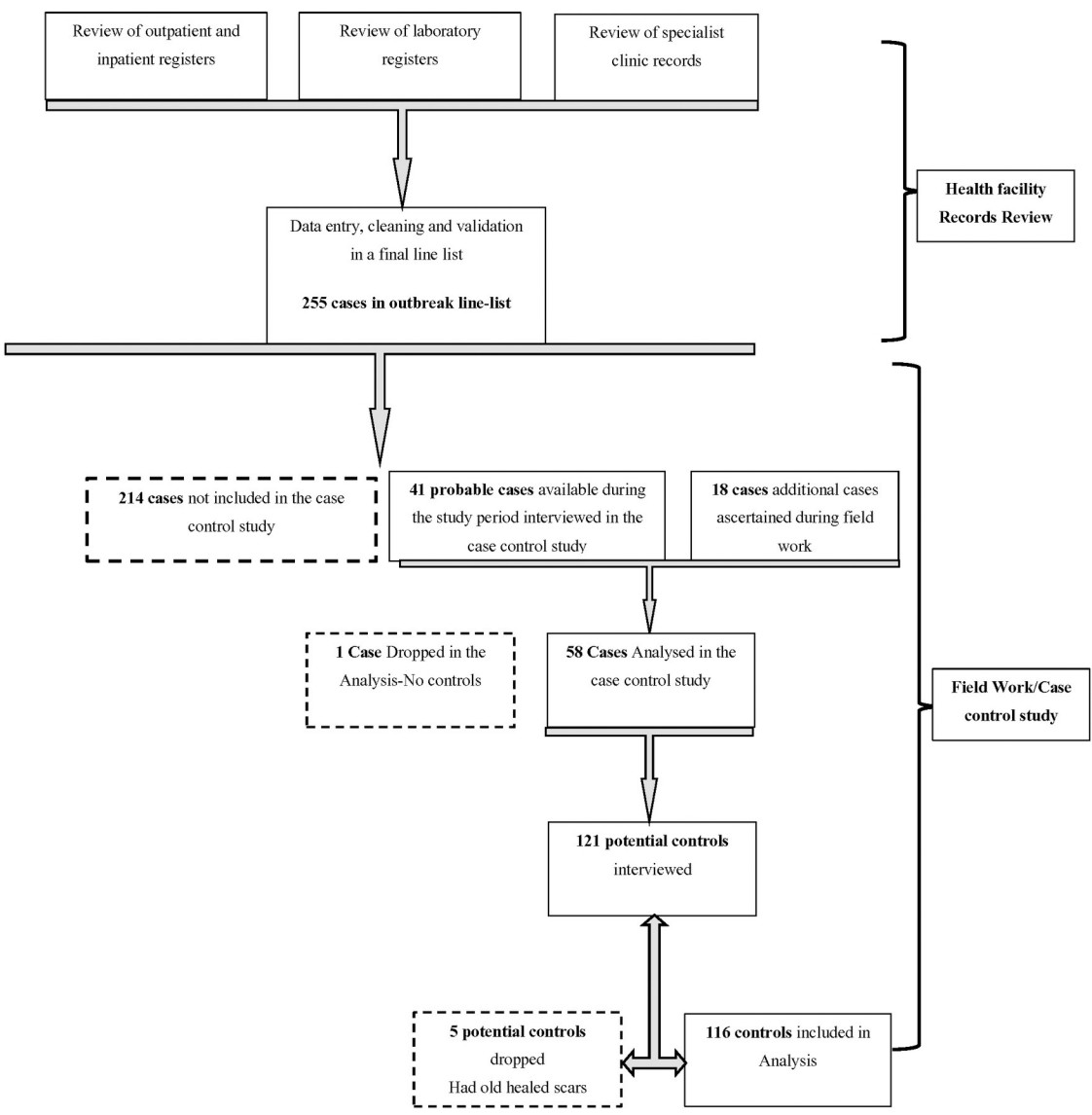

**Fig 2. Flow diagram of selection of cases and controls before and after field work.**

bivariate analysis were considered significant and were included in the 'group model' for each category of factors. To develop the final model, regression analysis was conducted using backward elimination method, starting with all factors that had a $P \leq 0.2$ in the group model to determine independent risk factors.

## Human subject's protection

This study was approved by Ministry of Health (MOH) in Kenya and was conducted as part of public health response to an acute event and as such was not reviewed by an ethical review body. Oral consent was obtained from the case-control study subjects and was documented in the study questionnaires. Study information was provided in form of written leaflets given to all study participants and displayed at strategic locations in health facilities in the study area (example in S1 Leaflet). An outbreak response protocol was developed and authorisation to

conduct the study was given by the MOH through the Field Epidemiology and Laboratory Training Program (Approval letter Ref No. MOH/DPH/DSRU/REG/07/Vol. 1) (letter in S1 Letter). Permission to conduct the study was also sought from the department of health in Nakuru county and respective health facilities where records were abstracted. Measures taken to assure confidentiality of the information provided during these interviews included conducting interviews in a private place convenient for respondents, storage of paper questionnaires in lockable cabinets accessible only with authorization of the principal investigator and password protection of de-identified data in electronic databases. Review of surveillance data and active case finding in the community were conducted as part of routine surveillance by the MOH, and all the data collected in the line-list was anonymized by dropping all personal identifiers (patient names, in-patient/out-patient numbers and phone contact) before analysis. Individuals who had active lesions at the time of the study were referred for free treatment at Nakuru county referral hospital

## Results

### Review of records

From the review of health facility records and house-to-house survey, a total of 255 cases of suspected CL were identified between 2010–2016. There was one death documented from a 67-year old woman on treatment for CL over the period of review. Of the identified cases, females constituted 48.6% (124/255) and the median age was 7 years (IQR 3–17). Cases occurred in clusters and up to 43% of cases originated from *Gitare* (73/255) and *Kambi-Turkana* (36/255) villages (**Table 1**).

Cases of suspected CL were recorded continuously throughout the period between 2010 and 2016 with occasional peaks in June of 2010 and December of 2014. Most (23.5%) cases of suspected CL were recorded in 2014 while the least (11.8%) cases were recorded in 2011. (**Fig 3**)

### Case control study

**Table 2** summarizes the demographic characteristics of cases and controls enrolled in the case control study. Males constituted 55.2% (32/58) of the cases and 49.1% (57/116) of the controls. Cases were predominantly young persons aged below 15 years (56.9%). The youngest case was 2 years while the oldest was 86 years old. There was significant difference in the distribution of cases and controls by type of occupation. However, education level was not significantly different between the two groups.

Among the cases, the median duration of illness was 2 years (range 1–4 years). Cases had multiple lesions with the majority (84.5% or 49/58) of cases presenting with both ulcerative and nodular lesions. The majority of lesions were located on the head and neck regions (81.6%, or 40/49) while 6.1% (3/49) were located in the hands and 2.0% (1/49) were in the foot. In thirteen cases (22.4%), both active ulcerative lesions and scars were found. Other symptoms observed among cases included pruiritus (15.5%), rash (5.2%), bruising (5.2%), skin infections (5.2%) and nasal stuffiness (3.4%). Various wound treatment remedies were cited by the cases including herbal medication (72.1%), skin ointments (41.9%) and antibiotics (25.6%).

In terms of household ownership and use of mosquito-nets, 10.3% (6/58) of the cases and 18.1% (21/116) of the controls owned and slept under a mosquito-net every night. Household indoor residual spraying was reported by less than 5% of both cases and controls. Sighting of wild mammals around homes was common among both cases and controls: Some 87.9% of cases and 57.8% of controls reported sighting rock hyraxes and 75.9% of the cases and 69.0% of the controls reported sighting mongooses near their homes. In terms of housing condition,

**Table 1. Demographic characteristics of suspected cases of cutaneous leishmaniasis based on review of records from nine health facilities in Gilgil, 2010–2016 (n = 255).**

| Cases by age group | Female | (%) | Male | (%) | Total | (%) |
|---|---|---|---|---|---|---|
| <5 years | 51 | (60.0) | 34 | (40.0) | 85 | (33.3) |
| 5–15 years | 38 | (43.7) | 49 | (56.3) | 87 | (34.1) |
| 16–34 years | 27 | (61.4) | 17 | (38.6) | 44 | (17.3) |
| 35–59 years | 9 | (81.8) | 2 | (18.2) | 11 | (4.3) |
| 60+ years | 1 | (20.0) | 4 | (80.0) | 5 | (2.0) |
| Age not recorded [a] | 12 | (52.2) | 11 | (47.2) | 23 | (9.0) |
| **Total** | **138** | **(54.1)** | **117** | **(45.9)** | **255** | **(100.0)** |
| **Cases by village of residence** | | | | | | |
| Gitare | 36 | (49.3) | 37 | (50.7) | 73 | (28.6) |
| Kambi Turkana | 21 | (58.3) | 15 | (41.7) | 36 | (14.1) |
| Oljorai | 9 | (50.0) | 9 | (50.0) | 18 | (7.1) |
| Kongasis | 11 | (68.8) | 5 | (31.3) | 16 | (6.3) |
| Diatomite | 8 | (66.7) | 4 | (33.3) | 12 | (4.7) |
| Other Villages | 59 | (57.8) | 43 | (42.2) | 102 | (40.0) |
| **Total** | **142** | **(55.7)** | **113** | **(44.3)** | **255** | **(100.0)** |

[a] Age variable not captured in hospital records

82.8% of cases and 97.4% of the controls lived in houses with roofing made of corrugated iron sheet, 56.9% of cases and 59.5% of the controls lived in houses with earthen floor, 27.6% of cases and 21.6% of the controls lived in houses with cracked floors. The majority of cases (91.4%) lived in houses located near a forest compared to 57.8% controls.

## Risk factor analysis

Potential risk factors were categorised into three groups for purposes of analysis: factors related to the individual, factors related to indoor dwelling environment and factors related to outdoor environment. Table 3 shows the distribution of cases and controls with corresponding crude and adjusted odds ratios for the variables analysed in the 3 'group models.

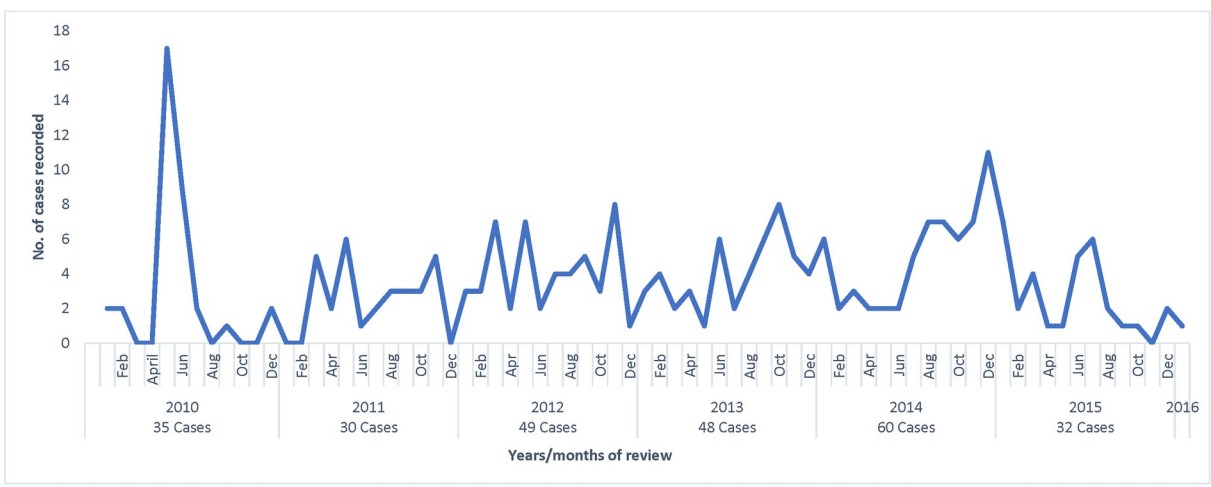

**Fig 3. Trends of patients treated for suspected cutaneous leishmaniasis in health facilities in Gilgil between January 2010 and January 2016 (N = 255).**

**Table 2. Demographic characteristics of cases and controls in Gilgil, Kenya 2016.**

| Variable | Cases | | Controls | | p |
|---|---|---|---|---|---|
| | (n = 58) | % | (n = 116) | % | |
| *Sex* | | | | | |
| Male | 32 | 55.2 | 57 | 49.1 | 0.451 |
| Female | 26 | 44.8 | 59 | 50.9 | |
| *Age Group* | | | | | |
| under 5 years | 7 | 12.1 | 12 | 10.3 | |
| 5–15 | 26 | 44.8 | 47 | 40.5 | |
| 16–34 | 6 | 10.3 | 24 | 20.7 | |
| 35–59 | 13 | 22.4 | 24 | 20.7 | |
| 60+ | 6 | 10.3 | 9 | 7.8 | 0.553 |
| *Level of education* | | | | | |
| Primary level and below | 55 | 94.8 | 103 | 88.8 | |
| Secondary level or more | 3 | 5.2 | 13 | 11.2 | 0.194 |
| *Occupation* | | | | | |
| In school | 20 | 34.5 | 51 | 44.0 | |
| Involving forest visit [b] | 20 | 34.5 | 12 | 10.3 | |
| Farming | 17 | 29.3 | 49 | 42.2 | |
| Other occupation [c] | 1 | 1.7 | 4 | 3.5 | 0.002 |

[b] **Occupation involving Forest visit**- included those who were engaged in charcoal burning, herding, hunting, stone masonry and mining.

[c] **Other occupation**- Included children out of school, housewives, those engaged in business and casual labourers.

In the first category, individuals who preferred staying outside their residence in the evening after sunset (OR 4.1, CI 1.2–16.2) and those whose primary occupation involved visiting forests (OR 4.6, CI 2.0–10.2) had significant associations with disease in the bivariate analysis. After adjusting for other factors in the multivariate model, only occupations involving forest visit remained significant with a reduced odds ratio of 3.8. Activities such as charcoal burning, hunting, herding, stone masonry and mining were included among the occupations involving forest visits. When assessed separately, these occupations had significantly large odds ratios, but this analysis is not reported here due to possible close link between each of these occupations with forest visits. Other individual attributes such as sex, level of education, history of travel or use of mosquito nets did not have any significant association with disease. **Table 3.**

Three of the five factors that were fitted in the second category of factors related to indoor transmission remained statistically significant in the multivariate analysis. This included sharing residence with a household member with typical skin lesions (OR 14.4, CI 3.8–79.3), residing in a house with alternative roofing materials (OR 7.9, CI 1.9–45.7) and residing in a house with cracked walls (OR 2.3, CI 1.0–4.9). These factors also showed significant associations with increased odds ratios after allowing for other factors in the multivariate model. **Table 3**

In the third category of factors related to outdoor environment, four of the eleven factors included in the analysis were significant: sighting rock hyraxes near residence (OR 5.3, CI 2.2–12.7), residing near a forest (OR 7.8, CI 2.8–26.4) and living close to a neighbour with typical skin lesion (OR 6.8, CI 2.8–16.0) had increased likelihood of CL. Three of these remained significant in the multivariate model but with reduced odds ratios. Having a cultivated crop farm surrounding the residence (OR 0.1, CI 0.0–0.4) appeared protective. This association remained significant in the multivariate model. **Table 3**

**Table 3. Analysis of the risk factors associated with cutaneous leishmaniasis in Gilgil-Kenya, 2016.**

| Variables | Cases N = 58 | | Controls N = 116 | | Crude OR (95% CI) | Adjusted OR (95% CI) | P value |
|---|---|---|---|---|---|---|---|
| | n | % | n | % | | | |
| **A. Factors related to the individual** | | | | | | | |
| Participant is male | 32 | 55.2 | 57 | 49.1 | 1.3 (0.7–2.4) | 1.0 (0.5–1.9) | 0.892 |
| Education level is primary level and below | 55 | 94.8 | 103 | 88.8 | 2.3 (0.6–13.1) | 2.5 (0.6–9.6) | 0.190 |
| Spending time outside home after sunset | 9 | 15.5 | 5 | 4.3 | 4.1 (1.2–16.2) | 2.1 (0.6–7.7) | 0.284 |
| Individual using bed net | 6 | 10.3 | 21 | 18.1 | 0.5 (0.2–1.5) | 0.6 (0.2–1.7) | 0.366 |
| Occupation involves forest visit [d] | 20 | 34.5 | 12 | 10.3 | 4.6 (2.0–10.2) | 3.8 (1.6–9.0) | 0.003 |
| History of travel | 9 | 18.4 | 15 | 12.9 | 1.2 (0.5–3.0) | 1.1 (0.4–2.9) | 0.916 |
| **B. Factors related to indoor dwelling environment** | | | | | | | |
| Roofing made of other materials [e] | 10 | 8.6 | 3 | 2.7 | 7.9 (1.9–45.7) | 13.6 (2.5–74.7) | 0.003 |
| Living in a house with cracked walls [f] | 48 | 82.8 | 79 | 68.1 | 2.3 (1.0–4.9) | 3.9 (1.1–13.6) | 0.032 |
| Living in a house with cracked floors [g] | 49 | 84.5 | 94 | 81.0 | 1.3 (0.5–3.0) | 1.4 (0.3–7.1) | 0.707 |
| 5 or fewer regular household members | 37 | 63.8 | 60 | 51.7 | 1.6 (0.9–3.1) | 1.8 (0.8–3.8) | 0.131 |
| Sharing residence with household member with ulcerating disease | 16 | 27.6 | 3 | 2.3 | 14.4 (3.8–79.3) | 16.0 (4.1–62.8) | <0.001 |
| **C. Factors related to outdoor environment** | | | | | | | |
| Sighting rock hyrax near residence | 51 | 87.9 | 67 | 57.8 | 5.3 (2.2–12.7) | 3.0 (0.9–9.3) | 0.065 |
| Sighting wild jackal near residence | 11 | 19.0 | 19 | 16.4 | 1.2 (0.5–2.7) | 1.2 (0.4–3.6) | 0.683 |
| Sighting porcupine near residence | 43 | 74.1 | 79 | 68.1 | 1.3 (0.7–2.8) | 0.5 (0.2–1.3) | 0.159 |
| Sighting mongoose near residence | 44 | 75.9 | 80 | 69.0 | 1.4 (0.7–2.9) | 1.1 (0.4–2.7) | 0.846 |
| Domestic dogs in the residence | 36 | 62.1 | 83 | 71.6 | 0.7 (0.3–1.3) | 1.3 (0.5–3.2) | 0.553 |
| Presence of a nearby forest or thicket | 53 | 91.4 | 67 | 57.8 | 7.8 (2.8–26.4) | 7.0 (2.0–24.7) | 0.003 |
| Presence of a nearby open water source [h] | 5 | 8.6 | 24 | 20.7 | 0.4 (0.1–1.1) | 0.4 (0.1–1.1) | 0.081 |
| Immediate neighbour has typical skin lesions [i] | 21 | 36.2 | 9 | 7.8 | 6.8 (2.8–16.0) | 3.1 (1.1–8.8) | 0.031 |
| Distant neighbour has typical skin lesions [i] | 17 | 29.3 | 41 | 35.3 | 0.8 (0.4–1.5) | 1.0 (0.4–2.4) | 0.937 |
| Presence of garbage mound near residence | | | | | 0.6 (0.3–1.1) | 0.6 (0.2–1.5) | 0.278 |
| **D. Protective factor** | | | | | | | |
| Presence of cultivated crop farm near residence | 46 | 79.3 | 113 | 97.4 | 0.1 (0.0–0.4) | 0.1 (0.0–0.5) | 0.006 |

[d] Occupation that involves forest visit included those who were engaged in charcoal burning, herding, hunting, stone masonry and mining.

[e] Other roofing materials included grass, leaves, earthen, wood or rocky caves.

[f] Cracked wall type included earthen walls, rocky caves or walls have visible cracks and crevices.

[g] Cracked floors included earthen, rocky caves, or floors with visible crevices.

[h] Open water sources included river, dam, well, bore hole, pond or spring.

[i] Typical skin lesions included a skin ulcer with typical raised edges and depressed centre or a skin plaque (a circumscribed, nodular or palpable skin lesion) on physical examination by a medical officer

In the final model, seven factors remained significant after controlling for all factors. Occupations that involve visit to the forest (aOR 3.4, CI 1.1–10.7), living in a house with cracked walls (aOR 5.5, CI 1.6–19.3), five or fewer household occupants (aOR 2.8, CI 1.1–7.1), sharing residence with a household member with typical ulcerating disease (aOR 26.7, CI 5.2–135.8), having of a forest in the neighbourhood of residence (aOR 5.8, CI 1.7–19.6) and having a neighbour with typical skin lesions (aOR 5.3, CI 1.8–15.7) were significant in the final model. Having a cultivated crop farm near the residence (aOR 0.1, CI 0.0–0.5) remained protective. The odds ratios for these associations are as shown in **Table 4**.

**Table 4. Final model showing adjusted odds ratios and 95% confidence intervals for factors related to cutaneous leishmaniasis in Gilgil-Kenya, 2016.**

| Variable | Adjusted odds Ratio | 95% CI | P-value |
|---|---|---|---|
| Occupation involves forest visit | 3.4 | 1.1–10.7 | 0.038 |
| Living in a house with cracked walls | 5.5 | 1.6–19.3 | 0.008 |
| Cultivated crop farm near residence | 0.1 | 0.0–0.5 | 0.007 |
| Presence of a nearby forest or thicket | 5.8 | 1.7–19.6 | 0.005 |
| Immediate neighbor with ulcerating disease | 5.3 | 1.8–15.7 | 0.003 |
| Sharing residence with household member with typical skin lesions | 26.7 | 5.2–135.8 | <0.001 |
| 5 or fewer regular household members | 2.8 | 1.1–7.1 | 0.029 |

## Discussion

### Review of hospital records

This study has highlighted the burden of CL in Gilgil through records review and house-to-house survey. Young children of school-going age were disproportionately affected, and cases mainly originated from two geographical locations. The study area is relatively sparsely populated and a finding of 255 cases in this area is significant. However, barring the quality of hospital records, the number of CL cases in this area could be higher. CL has been known to be present in this geographical area since the early 1990's [9,21]. It is therefore not surprising that we found evidence of continuous transmission of CL between 2010–2016. The official health reporting portal of Kenya MOH through the Kenya Health Information System (KHIS) platform does not incorporate CL among the monthly aggregate reports sent from health facilities across the country [22]. In this portal, reports of CL are ordinarily lumped and reported as "diseases of the skin" making it impossible to compute cases of CL at the national level. Coupled with inadequate diagnosis at health facility level, there is potential underreporting of CL over the period.

CL cases were clustered around two geographical locations in the study area: cases originated either from *Gitare*, the northern part of Gilgil along the rocky cliffs on the wall of the Great Rift Valley or from the southern end of Gilgil in *Útut* forest which is located in an area of solidified volcanic lava on the floor of the Great Rift Valley. Based on observations by the study team, both regions are remote, largely inaccessible, lack basic infrastructure and are inhabited by small scale farmers (*Gitare* Village) or forest dwellers (*Útut* Village). In *Gitare* village, crop farmers were observed to be encroaching the thickets around rocky escarpments near their homes for farmland and new homes were seen to be constructed in areas that were previously under forest cover. In *Útut* village, charcoal burning, stone mining, beekeeping and herding are majorly carried out in the forests by many residents.

The finding of clustering of cases in the two foci could point at accelerated CL transmission in these areas. Indeed, Sang et al described *Útut* forest, among other regions, as a focus of CL on the floor of the Great Rift Valley with the sand-flies also identified in this region [11]. This focalized transmission pattern could be attributed to three possible explanations. First, the sand-fly vector implicated in transmission of CL has a restricted flight radius, mostly flying within a range of 50 meters around their habitat and would therefore only bite and transmit CL in a localized geographical area [23]. Secondly, there is abundance of mammal reservoirs in these localities as evidenced by the increased likelihood (three-fold) of sighting of rock hyraxes by the majority of CL patients in the study area. Wild rodents including rock hyraxes have been described in literature as animal reservoirs of CL and could facilitate transmission of CL in this locality [11]. Lastly and importantly, these two localities epitomize high rate of human

encroachment to previously non-inhabited land. In *Gitare* village, crop farming is expanding while in *Útut* forest, more people are visiting the forest to burn charcoal, harvest honey, mine stones and herd livestock.

## Risk factors of cutaneous leishmaniasis

Through analysis of demographic, behavioural and environmental attributes, we have been able to identify independent risk factors that are associated with CL in the study area. Significant associations seen in analysis of risk factors in this study suggest an overlap of factors that promote the likelihood of occurrence of CL at individual, indoor or outdoor levels. Individual factors included behavioural (spending time outside residence in the evening after sunset) and occupational factors (involving visit to the forest). Indoor risk factors included houses with cracked walls, households with fewer inhabitants and households with at least one member infected with CL. Outdoor transmission was associated with individuals residing close to a forest, close to a neighbour with CL or individuals who sighted wild rodents such as the rock hyrax near their residence.

The sand-fly vector for CL is described in literature as naturally anthropophilic and crepuscular, preferring to bite both indoors and outdoors in the evenings and early mornings [4,12,24]. In previous studies, the sand-fly has been identified in parts on Kenya, including Gilgil [12,23,25,26]. Among the indoor factors, we observed almost four-fold increased likelihood of CL among residents of houses with cracked walls. It is possible that large cracks and crevices on walls of residences provided daytime hiding environment for the sand-fly after a blood meal. Perhaps a more direct evidence of indoor transmission is supported by our finding of 16-fold increased risk of CL in households where at least one member had typical ulcers. This observation could be explained in part by familial clustering tendencies of CL that have been reported in previous studies [27–29] or by a possible presence of high sand-fly vector density within these residential units [4].

Our finding of increased risk of CL in households with 5 or less inhabitants marks a departure from what has been observed in most studies since a large house-hold size (number of regular residents of a household) and high population density are considered as proxy indicators of poverty which has in turn been associated with CL [2]. One possible explanation for this reverse association could be that small household sizes could potentially predispose the few household occupants to frequent sand-fly-human contact through bites [4]. Such an inverse relationship has also been observed in the case of common arthropod-vector borne diseases such as malaria [30,31].

Presence of forest near residences and residents engaged in occupations that involved visit to the forests including charcoal burning, herding, hunting, wild honey harvesting and stone mining, all had increased likelihood of CL. Forest and thickets are likely to provide suitable habitats to immature and mature forms of the sand-fly vector. Additionally, the rocky caves and thickets around Gilgil have also been known to be infested with hyraxes and other mammals that are natural reservoirs of CL, and were likely to be sighted around homes among those with CL [21,32]. As a result of increased population pressure, changes in land use (from forestry to agriculture) and deforestation, reservoir mammals have been known to migrate closer to human residences [2,4,24]. Therefore, one plausible explanation for increased risk observed among residents who frequent forests for livelihood could be that such exposures would increase the frequency and duration of sand-fly-human contact. It appears that frequent vector-human contact either due to increased density of sand-fly in the human dwelling environment or as a result of humans venturing in sand-fly infested habitats in the forests could

explain the increased risk of transmission in both indoor and outdoor settings, with the constant being increased exposure to the vector.

Studies have consistently shown more cases of CL among poor, neglected populations who are likely to be less educated and mostly unemployed [2,33]. In our study, most of the sampled population (94.8%) had primary level of education or less, 20% were unemployed and relied on menial jobs to occupy their time and generate income and 72.1% used herbal medication for treatment of skin lesions. All these findings reinforce this pattern of association. Other publications have also shown similar findings [34,35]. However, analysis of level of education and employment status did not show significant associations in the bivariate analysis in our study.

As expected, cases were younger, a finding that is consistent with available literature that has shown that prevalence of CL increases with age in from early childhood then levels off by 15 years of age [4,34]. Even though this observation has been explained by a gradual acquisition of immunity among susceptible persons with increasing age, the explanation that appears more plausible is that as children grow up and become more mobile, they are likely to be exposed to bites when they visit forests around their homes. Perhaps, an interplay of these two factors would be very likely. Some cases had both active lesions (typical ulcer) and old scars at various stages of healing. This could imply repeated infections in the same individual, a possible indication that immunity developed following primary infection could not likely be life-long as seen elsewhere in literature [36].

Skin ulcers were frequently located in head and neck regions of the body and less frequently in hands and feet. Similar findings have been recorded in other studies [11,37]. The wounds mainly affected exposed areas of the body which are commonly bitten by the sand-fly. Mouth and nostril ulcers were also observed among some of the cases, possibly because such areas of the body are also exposed to insect bites. Ordinarily, mucosal lesions are typical of muco-cutaneous leishmaniasis (MCL), however, MCL would be highly unlikely to occur in this region given that the agent (*L. tropica*) that is known to be present in this area is mainly associated with CL [11].

Despite its strengths, this study had some limitations. Hospital records available for the review were either incomplete or inaccurate. Lack of a proper hospital records could lead to underestimating/overestimating the burden of CL. Secondly, laboratory confirmation was not done for the identified cases. Therefore, some of the chosen controls could actually be incubating cases or false positives. Thirdly, it was also not possible to compute the incidence of disease as some of the cases enlisted reported multiple infections over time. Moreover, CL has a long latency period hence some cases identified in the hospital records could be prevalent cases as opposed to incident cases. Owing to the long duration of disease, there is potential for recall bias could affect the quality of responses that we got from respondents. Despite our finding of strong association between individual, indoor and peri-domestic factors in the transmission of CL, our study did not estimate the relative contribution of each of these factors in transmission of CL.

## Conclusion and recommendations

This study has highlighted the burden of CL in Gilgil. However, due to sparse data in the visited facilities, the true burden of disease could be higher. Cases were reported throughout the years, consistent with a locally endemic disease transmitted continuously throughout the period. This study highlighted indoor and outdoor risk factors that promoted clustering of cases among household members and focalized transmission pattern in specific neighbourhoods in the study area. Occupations and activities that involve visiting forests, residing near forests or sharing a house or neighbourhood with a person who has CL were identified as

significant exposures of the disease. CL lesions mostly affected younger or older residents with lesions mainly located in exposed parts of the body.

There is need to strengthen diagnosis and reporting for CL in Gilgil in order to provide a better estimate of the disease burden. Emphasis should be put on quality of data collected in the affected health facilities. Tailored control interventions including indoor residual spraying, barrier methods (such as insecticide treated nets, insect repellents and wearing protective clothing) and destruction of vector breeding grounds would be effective when deployed in this area given the focalized pattern of transmission of CL driven by human interaction with known sand-fly hotspots such as forests. However, owing to the anticipated difficulties in deployment of these control measures, we recommend that studies be conducted to establish the feasibility and effectiveness of these interventions among residents of Gilgil. Modification of human behaviour in areas of known transmission risks would appear to be potentially effective as a disease control strategy. Discouraging practices such as hunting and charcoal burning would in theory be effective but would not be practical without alternative sources of income. However, the role of environmental factors and wild mammals in disease transmission should be investigated further.

## Supporting information

**S1 Checklist. STROBE checklist.**
(DOC)

**S1 Questionnaire. Cutaneous leishmaniasis questionnaire.**
(PDF)

**S1 Leaflet. Cutaneous leishmaniasis information leaflet.**
(PDF)

**S1 Letter. Ministry of Health (MOH) approval letter.**
(PDF)

**S1 Data. Cutaneous leishmaniasis dataset.**
(XLSX)

## Acknowledgments

I would like to acknowledge the contribution of the following individuals and institutions for their various contributions in this work: Dr Joe Lenai, Mr Gerald W. Maina and the Nakuru county health management team, Dr Ester Kanyiru and the entire Gigil sub-county health management team, International Livestock Research Institute (Kenya), FELTP Faculty (Kenya), Washington State University Global Health programs Kenya (WSU-GH Kenya) and the Unit for Neglected Tropical Diseases in Kenya. The findings and conclusions in this paper are those of the author and do not necessarily reflect the official views of the supporting institutions.

## Author Contributions

**Conceptualization:** Isaac Ngere, Waqo Gufu Boru, Abdikadir Isack, Joshua Muiruri, Mark Obonyo, Sultani Matendechero, Zeinab Gura.

**Data curation:** Isaac Ngere, Abdikadir Isack, Joshua Muiruri, Mark Obonyo, Sultani Matendechero, Zeinab Gura.

**Formal analysis:** Isaac Ngere, Waqo Gufu Boru, Abdikadir Isack, Joshua Muiruri, Mark Obonyo, Sultani Matendechero, Zeinab Gura.

**Funding acquisition:** Isaac Ngere, Waqo Gufu Boru, Sultani Matendechero, Zeinab Gura.

**Investigation:** Isaac Ngere, Waqo Gufu Boru, Abdikadir Isack, Joshua Muiruri, Mark Obonyo, Sultani Matendechero.

**Methodology:** Isaac Ngere, Abdikadir Isack, Joshua Muiruri, Mark Obonyo.

**Project administration:** Isaac Ngere, Sultani Matendechero, Zeinab Gura.

**Resources:** Isaac Ngere, Waqo Gufu Boru, Sultani Matendechero, Zeinab Gura.

**Software:** Isaac Ngere.

**Supervision:** Isaac Ngere, Waqo Gufu Boru, Mark Obonyo, Sultani Matendechero, Zeinab Gura.

**Validation:** Isaac Ngere, Mark Obonyo, Zeinab Gura.

**Visualization:** Isaac Ngere.

**Writing – original draft:** Isaac Ngere.

**Writing – review & editing:** Isaac Ngere, Waqo Gufu Boru, Abdikadir Isack, Mark Obonyo, Sultani Matendechero, Zeinab Gura.

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
