## [Decision Letter · Decision Letter 0]

5 Sep 2019

PONE-D-19-19202

Cases of Cutaneous Leishmaniasis in a peri-urban settlement in Kenya, 2016

PLOS ONE

Dear Dr Ngere,

Thank you very much for submitting your manuscript "Cases of Cutaneous Leishmaniasis in a peri-urban settlement in Kenya, 2016" (#PONE-D-19-19202) for review by PLOS ONE. As with all papers submitted to the journal, your manuscript was fully evaluated by academic editor (myself) and by independent peer reviewers. The reviewers appreciated the attention to an important health topic, but they raised substantial concerns about the paper that must be addressed before this manuscript can be accurately assessed for meeting the PLOS ONE criteria. Therefore, if you feel these issues can be adequately addressed, we invite you to submit a revised version of the manuscript that addresses the points raised during the review process. We can’t, of course, promise publication at that time.

We would appreciate receiving your revised manuscript by Oct 20 2019 11:59PM. To enhance the reproducibility of your results, we recommend that if applicable you deposit your laboratory protocols in protocols.io, where a protocol can be assigned its own identifier (DOI) such that it can be cited independently in the future. For instructions see: http://journals.plos.org/plosone/s/submission-guidelines#loc-laboratory-protocols

We look forward to receiving your revised manuscript.

Kind regards,

Abdallah M. Samy, PhD

Academic Editor

PLOS ONE

**Additional Editor Comments:**

I invited and received four reviews for your manuscript. All reviews raised some substantial concerns about your manuscript as it currently stands. I read through their comments and found that they coincided on several points, and that their reviews were uniformly solid and detailed. I read the manuscript myself, and I must say that I coincide with the reviewers' points entirely. As such, I would recommend “major revision”. I would kindly ask you to go through all comments raised by each reviewer and address them properly before sending a revised version of this manuscript. Please check all PLOS ONE style requirements available via https://journals.plos.org/plosone/s/submission-guidelines before submitting the revised version. Finally, please consider a careful language revision for your manuscript.

2. Please include additional information regarding the survey or questionnaire used in the study and ensure that you have provided sufficient details that others could replicate the analyses. For instance, if you developed a questionnaire as part of this study and it is not under a copyright more restrictive than CC-BY, please include a copy, in both the original language and English, as Supporting Information. Moreover, please include more details on how the questionnaire was pre-tested, and whether it was validated.

3. Please correct your reference to "p=0.000" to "p<0.001" or as similarly appropriate, as p values cannot equal zero.

**Reviewers' comments:**

Reviewer's Responses to Questions

**Comments to the Author**

1. Is the manuscript technically sound, and do the data support the conclusions?

Reviewer #1: No

Reviewer #2: Partly

Reviewer #3: Yes

Reviewer #4: Partly

2. Has the statistical analysis been performed appropriately and rigorously? 

Reviewer #1: Yes

Reviewer #2: I Don't Know

Reviewer #3: Yes

Reviewer #4: I Don't Know

3. Have the authors made all data underlying the findings in their manuscript fully available?

Reviewer #1: No

Reviewer #2: Yes

Reviewer #3: No

Reviewer #4: No

4. Is the manuscript presented in an intelligible fashion and written in standard English?

Reviewer #1: No

Reviewer #2: Yes

Reviewer #3: Yes

Reviewer #4: Yes

5. Review Comments to the Author

Reviewer #1: The efforts of the authors are appreciated. However, there is a grave concern about the cases recruited in the study. Suspected cases, with the definition authors provided, must not be included in the study.

Reviewer #2: Cases of Cutaneous Leishmaniasis in a peri-urban settlement in Kenya, 2016

Ngere, et al.

1. The time period the clinical reports that were reviewed covered 2010-2016. This is clearly indicated in the paper text but should be indicated in the initial paper Summary Section as well.

2. Ethical review of the study was not required as this was a MOH response to an acute situation. It seems however, that the appropriate measures were taken to assure patient protections (confidentiality and consent). Was MOH number (approval) given to the Study—might this be cited?

3. The authors state that cases are higher in June-December period. However, this is really only the case for 2014. This may be more a case of continuous transmission. It is not clear why there is a peak for June 2010 and then little disease reported subsequently. In 2011, 2012 and even 2013- disease appears continuous.

4. In reporting factors relevant to disease—some are positively related (forest time, neighbors with disease) while others are negatively associated (protective- crop growing). These are not really distinguished in the tables—although the text is clear. It would be useful to make this clearer in the Tables presented.

5. It is unclear to this reviewer that some of the hypotheses concerning the epidemiological features of the leishmaniasis outbreak are reasonable. 1) The authors suggest (Discussion Section) that as many lesions are on the nose and mouth that mucocutaneous leishmaniasis may be occurring. This would be unusual for the species likely transmitting in the area- which is likely (based on the reservoir hosts implicated), L. tropica. Actually a report in 1994 (Trans R Soc Trop Med Hyg. 1994 . 88:35-7; note this reference (9) has been miscited in the Reference Section) from this area indicated that this species is in circulation in this area of Kenya. No mention is made of this point though other findings of this paper were mentioned in the text. MCL is not straightforward (as the authors appear to realize) and these really sound more like these lesions are where the insect bite occurred and not really a dissemination/metastasis from a cutaneous site to mucosal tissue. L. tropica (to date) is most commonly associated with CL rather than MCL. 2) The authors (Discussion) mention that there is a higher susceptibility at the ends of the age spectrum – older and younger. They suggest that the susceptibilities of these populations might reflect impaired immune responses associated with these groups. However, this does not match their data. The lower immune responses of children are generally associated with infants and the group here with higher level of infection was the >5-15 year olds and not the <5 year old group. In fact the highest incidence of cases (48% of total) is in this group. Few cases appear in the >60 years group. This group (5-15 years) would likely be more mobile and in and out of the nearby forests. Certainly, the data available could be used to examine this point. However, immune impairment does not seem reasonable. 3) In Table 4, it is indicated that one of the “interior” factors for disease is “Living in a house with cracked walls”. This sounds like the barrier to the outside is broken. Would this not reflect the exterior conditions then? In these cases, were the houses closer to the forests (another disease factor) as well? It would be useful to assess this in terms of identifying intervenable control measures.

Minor Points:

1. On the Title Page- it is not clear what the symbol & represents.

2. Line 114- Leishmania donovani, the d should not be capitalized; line 115, the same for the c in L. chagasi. The Word program is annoying in that it does this- capitalizes letters after periods (.).

3. Map in Figure 1 is useful but could be improved with the addition of specific geographical landmarks (City, river).

Reviewer #3: First

The study is within the scope of the journal considered about one of the neglected diseases in the world.

The title should be matched the objective, methodology and conclusion, it better to be as

“Risk factors of cutaneous leishmaniasis in a pre-urban settlement in Kenia, 2016”

Abstract should not include any abbreviations and not exceed more than 300 wards. So the abbreviation “CL” should be dropped from the conclusion section of the abstract and the abstract wards (389) should be decreased to match the journal criteria.

Really, the study has a clear design of a case control study type however, all the cases of the study group either suspected or probably cases of cutaneous leishmaniasis. All cases diagnosed clinically so in conclusion section, it is better to use “there is evidence” rather than “there is strong evidence”

second

The pronoun “we” was used several time through the paper, it is better to avoid using the subjective pronouns and it should be dropped from the text.

-In the line number 93, the abbreviation “CL” was not previously defined and should be written as cutaneous leishmaniasis rather than the short form “CL”.

-In the line number 81, “intermediate hosted” it is better to be “reservoir” rather than intermediate.

-In the line number 182, drop the wards “at the” from the sentence

-In the line number 228, the ward “while” it is better to be dropped.

-In the line number 262, it is better to add word “Kenia” at the end of the title of table 3.

-In the line number 290, it is better to add the time of the study “2016” at the end of the title of the table 4.

-In the line, number 337, it is better to uses “reservoir mammals” rather than “infected wild rodents”

-In the line number 474, the references 15 should be corrected by 17

-In the line number 481, the reference 18 should be corrected number 20

Finlay

The map should be clearer illustrating all valleys, villages and cities which were mentioned through the paper.

Reviewer #4: It is a well written manuscript that provides further information on CL in Kenya, which is an understudied topic. However, there are some important issues in the methodology (e.g. definition of cases, how matching was done) and I also have my concerns about the ethical section. More details are provided below.

General

Abbreviations should be used consistently. CL and cutaneous leishmaniasis are used throughout the manuscript (e.g. in full in line 293). After the first time (line 70?) using the full name the abbreviation should be used.

Introduction

Line 93, there are several vectors for CL, which one is meant here?

Comments methods

General

To make what was done easier to understand the paper should consequently discuss the hospital review first (and separately) and the case control study after.

Study site:

The total catchment area of the health facilities should be indicated. Is 255 cases a lot? Understanding the context will help judge that.

The health facilities in Morendat and Mbaruk should be described in more detail.

Why was the gilgil sub-county chosen?

Line 113-114: which sandfly is meant? Did they really identify Leishmania infantum in Kenya? The agents described here seem wrong, especially for CL in Kenya.

Study population:

As a whole, the selection of the study population should be more clearly described. Specifically, the relationship between the line list made based on the hospital records to the case control study. From the 255 only 41 were included as cases. How were the 41 selected? What happened to the rest? Could there have been a bias in selecting these 41? Also for the controls, did all the controls agree to participate in the study?

How was the house to house survey used to detect the additional 18 cases carried out? What was the strategy? What were the exact criteria to define cases in the house to house survey.

There is mention of suspected, probable or confirmed cases, and that all are included as cases. However, is this correct? How many were actually suspected, probably and confirmed?

Were there any patients with missing data? Did this lead to any exclusions?

Definition of cases and control:

I have a big issue with the definition of cases, as I fear the definition of a CL case is too broad, and therefore the cases contain false positives. Patients with ‘dermatitis’, ‘skin infection’, or ‘skin wound’ now all seem to be recorded as cases, which does not seem correct. This may severely impact the study.

On line 128 it also states that a probable case was defined as a patient with a typical ulcer (who determines what is a typical ulcer) or plaque, ascertained by a medical officer in the study team during the study period. However, the study period was only 2 weeks, so how can this refer to the part of the study that led to the line list (which to my understanding is based on retrospective file review).

In addition, why are only ulcers or plaques mentioned as probable CL, while Sang already stated in 1993 that lesions were rarely ulcerated, and Toroitich also said that many lesions were in fact nodular.

-Why was no laboratory confirmation done? Logistical challenges are mentioned, these should be specified.

-For the record review, how experienced were the staff in the health facilities in diagnosing CL?

- How experienced was the medical officer

Field work

It states that a lab scientist was there during the field visit, but if no lab confirmation was done, what did this person actually do?

Matching

Why was matching done by neighbourhood done? This does not make sense when factors related to the outdoor environment are one of the main interests of the study. By matching by neighbourhood, neighbourhood itself (micro-environment) cannot be studied properly as a risk factor in that way. �I think the third level of factors related to outdoor environment cannot be properly studied if neighbours are chosen as controls (unless the houses are very far apart, which should then be adequately explained).

Because no direct neighbours of ‘CL cases’ could be taken as controls, how can having an immediate neighbour with CL be studied as a risk factor?

Why matching when multivariable regression is done in analysis to deal with confounding anyway?

Sample size

The manuscript states that OpenEpi was used to calculate the sample size. However, OpenEpi does not seem to have an option for calculating sample size for matched case control studies. When I redo the sample size calculation in open epi using unmatched case control I get 180, 167 or 189, not 174. It should be clearly indicated how they reached 174. After trying to repeat the sample size calulcation it seems that a sample size calculation for matched case control assuming correlation is 0.15 was used. However, this is not explained clearly, and if 0.15 correlation was used, this should be explained.

Sample size uses assumption that mosquito net use was present in 31% of controls. In fact this was <20%. I think this may lead to your study being underpowered, as when I retrospectively calculate the sample size using the 18% proportion controls using a mosquito net, the sample size is 306, 279 or 317, which is almost double of the current sample size.

Ethics:

This study was not reviewed by any ethical review body. The reason given is that the study was conducted as part of an MOH led public health response to an acute event. However, to my understanding even research carried out in emergency response (e.g. in Ebola research) requires (expedited) ethical review. The researchers should elaborate more on the reason and regulations as to why the study was not reviewed by any ethical board. Just because permission was given by the health authorities does not automatically mean the study is carried out according to the ethical principles of medical research. For instance regarding the consent procedures (see below).

The manuscript states that oral consent was obtained. However, there is no mention of any proof of this consent (e.g. recording of oral consent, or ideally written consent). It is also not stated clearly why for this study oral (non-recorded) consent suffices. I have my doubts about this, and therefore the reasons for these matters should be explicitly explained.

In addition, the informed consent should be explained in more detail (especially since it was not approved by any IRB), was the consent procedure organised in an appropriate and complete manner? Did the patient receive a participant information sheet describing the research and giving the contact information of the PI?

Measures were taken to assure confidentiality. Please specify which methods.

The manuscript states that no personal identifying information was collected or analysed. However, in line 133 it states that name, sex, age, date seen at the facility, residence, signs and symptoms, diagnosis and treatment given were all among collected data. This would definitely qualify as personal identifying information.

Why use the 3 steps in multivariate? Why not skip the analysis per level?

Results

It is mentioned that one death was suspected due to CL. This is strange as CL is normally not deadly, and therefore this should be elaborated on.

Line 203: IQR should give the 25th and 75th percentile, and not just one value)

Line 218-219 should be left out, as matching means that the age is very similar

Line 220: states that distribution of other social and demographic variables studied among cases and controls was comparable. However, occupation was significantly different between the two groups.

Line 227: mean duration of illness was two years .

228: it says all cases had typical ulcers, but it does not make sense to describe a feature that was used to select the population. If typical ulcers were the way the population was selected, of course all patients will have typical ulcers.

The number of lesions should be mentioned. If the big majority of cases had both a typical ulcer but also a papule or nodule, does this mean they had multiple lesions with different presentations or one lesion with both an ulcerative and indurated presentation?

Line 232: why mention nasal stuffiness? This seems random, unless you want to link it MCL.

Line 238-244. If patients were close neighbours how can there be a difference in sighting rock hyraxes around the house or in living close to the forest? Could this be purely due to bias?

252: here 2 decimals are given, throughout the manuscript it should be consistent, either 1 or two decimals for percentages and CI.

Line 257: certain activities were not analysed because they could potentially have a close link with forest visits. However they should be analysed in multivariable which can check whether they are independent risk factors.

Are the factors displayed in table 3 the only ones that were assessed? Or did a selection take place and only certain ones are displayed? This should be described.

Table 2 and 3 can be merged into one, as there is significant overlap.

Line 264: should bivariate analysis not be multivariate here?

The results from the risk factor analysis can be shortened. The essence should be which factors were risk factors in the bivariate, and how many of them were independent risk factors after multivariate analysis?

Discussion

I feel the burden of CL has not been sufficiently highlighted. At least, more background should be given that can help quantify the burden in relation to the study area.

Limitations regarding recall bias and the case definition should be mentioned.

Line 310: this seems to be over interpretation of the findings, this study does not allow saying anything about indoor or outdoor transmission being observed.

The CL outbreak of 2016 is never mentioned again. Was this actually an outbreak?

In line 355 it mentioned there may be accelerated transmission. A reference from 1993 is given. This indicates that CL is not a new thing here and therefore points in the other direction in fact.

Line 369: it is not clear whether the statement of low levels of education is about the cases in the case control study, the hospital records or in the population comprising both cases and controls. In addition the 90.8% that is mentioned to have less than primary education can not be found in any of the tables, which mention 94.8% of cases having primary education or less (not less than primary education).

Line 375: patients above 35 are mentioned being ‘older individuals.’ This statement is related to immunosuppression of old age. This is not appropriate, as only 10% of patients were above 60 in fact.

Line 386: states the most common presentation of cases was a skin ulcer, if this was the way the cases were selected do not use it to describe the presenting feature of CL.

The limitations of the case control study should be elaborated on, as there are quite a few (see comments from methods section).

Conclusions

404: no real grounds for this statement. What is large?

407: study does not provide real evidence on transmission, just on risk factors

414: what is meant by ‘there is need to institute surveillance for CL’?

References

Please correct the mistakes here, I found a few

-ref 9: should be Sang D, Ashford RW, Njeru WK etc.

-ref 10: authors name is Toroitich AK (not Itor Toroitich)

-ref 11: third author’s first rather than last name is written in full

6. PLOS authors have the option to publish the peer review history of their article (what does this mean?). If published, this will include your full peer review and any attached files.

Reviewer #1: No

Reviewer #2: No

Reviewer #3: Yes: Mohammed Musid Alkulaibi

Reviewer #4: No

---

## [Author Response · Author response to Decision Letter 0]

31 Oct 2019

Dear Editor,

The following are the responses to comments raised by reviewers on my manuscript “Cases of Cutaneous Leishmaniasis in a peri-urban settlement in Kenya, 2016" (#PONE-D-19-19202):

Editor comments

1. Please check all PLOS ONE style requirements available via https://journals.plos.org/plosone/s/submission-guidelines before submitting the revised version. Finally, please consider a careful language revision for your manuscript

These have been addressed. I have revised sections of the manuscript in line with the suggested changes from the reviewers. I have aligned the document to PLS ONE style requirements.

2. Additional comments (journal requirements): 

a. Please ensure that your manuscript meets PLOS ONE's style requirements, including those for file naming

This has been addressed

b. Please include additional information regarding the survey or questionnaire used in the study and ensure that you have provided sufficient details that others could replicate the analyses. For instance, if you developed a questionnaire as part of this study and it is not under a copyright more restrictive than CC-BY, please include a copy, in both the original language and English, as Supporting Information. Moreover, please include more details on how the questionnaire was pre-tested, and whether it was validated

The study questionnaire has been included as supporting information S2

c. Please correct your reference to "p=0.000" to "p<0.001" or as similarly appropriate, as p values cannot equal zero

These changes have been made in the revised manuscript.

d. Data availability and sharing: 

There is no ethical or legal restriction against sharing data. The minimal anonymized dataset for both the records review and case-control study have been shared in as supporting information file (“S5_Cutaneos_Leishmaniasis_Data”)

Reviewer #1: 

1. The efforts of the authors are appreciated. However, there is a grave concern about the cases recruited in the study. Suspected cases, with the definition authors provided, must not be included in the study

‘Suspected cases’ were only included in the review of hospital records. Only probable cases were included in the case control study. Probable cases were defined as residents with typical ulcers (with typical raised edges and depressed center) or plaques (a circumscribed palpable or visible lesion made up of multiple coalescent nodules or papules) as was ascertained by an experienced medical officer in the study team.

Reviewer #2:

2. The time period the clinical reports that were reviewed covered 2010-2016. This is clearly indicated in the paper text but should be indicated in the initial paper Summary Section as well.

This has been addressed (See revised manuscript abstract section)

3. Ethical review of the study was not required as this was a MOH response to an acute situation. It seems however, that the appropriate measures were taken to assure patient protections (confidentiality and consent). Was MOH number (approval) given to the Study—might this be cited?

MOH approval was given for the outbreak response, see the approval letter in the supplementary materials.

4. The authors state that cases are higher in June-December period. However, this is really only the case for 2014. This may be more a case of continuous transmission. It is not clear why there is a peak for June 2010 and then little disease reported subsequently. In 2011, 2012 and even 2013- disease appears continuous.

This has been addressed. Continuous transmission was seen throughout the period with occasional peaks in June 2010 and December 2014

5. In reporting factors relevant to disease—some are positively related (forest time, neighbors with disease) while others are negatively associated (protective- crop growing). These are not really distinguished in the tables—although the text is clear. It would be useful to make this clearer in the Tables presented.

I have revised Table 3 to show protective factors in a separate category

6. It is unclear to this reviewer that some of the hypotheses concerning the epidemiological features of the leishmaniasis outbreak are reasonable. 1) The authors suggest (Discussion Section) that as many lesions are on the nose and mouth that mucocutaneous leishmaniasis may be occurring. This would be unusual for the species likely transmitting in the area- which is likely (based on the reservoir hosts implicated), L. tropica. Actually a report in 1994 (Trans R Soc Trop Med Hyg. 1994 . 88:35-7; note this reference (9) has been miscited in the Reference Section) from this area indicated that this species is in circulation in this area of Kenya. No mention is made of this point though other findings of this paper were mentioned in the text. MCL is not straightforward (as the authors appear to realize) and these really sound more like these lesions are where the insect bite occurred and not really a dissemination/metastasis from a cutaneous site to mucosal tissue. L. tropica (to date) is most commonly associated with CL rather than MCL.

After carefully reviewing available references and discussion with subject matter expert, this statement has been revised to read “Mouth and nostril ulcers were observed among some of the cases, possibly because such areas of the body are also exposed to insect bites. Ordinarily, mucosal lesions are typical of muco-cutaneous leishmaniasis (MCL), however, MCL would be highly unlikely to occur in this region given that the agent (L. tropica) that is known to be present in this area is mainly associated with CL”- See text in the revised manuscript

7. The authors (Discussion) mention that there is a higher susceptibility at the ends of the age spectrum – older and younger. They suggest that the susceptibilities of these populations might reflect impaired immune responses associated with these groups. However, this does not match their data. The lower immune responses of children are generally associated with infants and the group here with higher level of infection was the >5-15 year olds and not the <5 year old group. In fact the highest incidence of cases (48% of total) is in this group. Few cases appear in the >60 years group. This group (5-15 years) would likely be more mobile and in and out of the nearby forests. Certainly, the data available could be used to examine this point. However, immune impairment does not seem reasonable.

Indeed, our data does show that more cases of CL were younger. Some literature explains that prevalence of CL increases with age up to about 15 years after immunity is acquired following childhood infections in endemic areas (Reithinger et al 2003). However, it is also likely that as children grow and become more mobile, they get exposed many times when they visit forest around their homes. We have therefore factored in these explanations in the revised discussion section of our manuscript.

8. In Table 4, it is indicated that one of the “interior” factors for disease is “Living in a house with cracked walls”. This sounds like the barrier to the outside is broken. Would this not reflect the exterior conditions then? In these cases, were the houses closer to the forests (another disease factor) as well? It would be useful to assess this in terms of identifying intervenable control measures.

Cracked walls had visible lines, sometimes with crevices but not broken apart, therefore, these constituted ‘indoor factors’. Most of these walls were made of mud/earthen and rocks. These cracks provided daytime hiding places for the sand-fly after blood meal.

9. Minor Points:

a. On the Title Page- it is not clear what the symbol & represents

‘’ is an ampersand symbol, and is used to represent the 2nd set of equal contributors in the authors list. The symbol is recommended in the PLOS ONE formatting guidelines https://journals.plos.org/plosone/s/file?id=ba62/PLOSOne_formatting_sample_title_authors_affiliations.pdf

b. Line 114- Leishmania donovani, the d should not be capitalized; line 115, the same for the c in L. chagasi. The Word program is annoying in that it does this- capitalizes letters after periods (.)

Corrected

c. Map in Figure 1 is useful but could be improved with the addition of specific geographical landmarks (City, river).

Corrected (Fig. 1)

Reviewer #3:

1. The study is within the scope of the journal considered about one of the neglected diseases in the world. The title should be matched the objective, methodology and conclusion, it better to be as. “Risk factors of cutaneous leishmaniasis in a pre-urban settlement in Kenia, 2016”

I have revised the title to read “Burden and Risk Factors of Cutaneous Leishmaniasis in a peri-urban settlement in Kenya, 2016” in line with the major objectives of the study. This study identified and characterized cases of cutaneous leishmaniasis in Gilgil through a review of records and active case search in the community to establish the burden of cutaneous leishmaniasis and explored individual, indoor and outdoor factors that may promote spread of the disease. 

2. Abstract should not include any abbreviations and not exceed more than 300 wards. So the abbreviation “CL” should be dropped from the conclusion section of the abstract and the abstract wards (389) should be decreased to match the journal criteria.

The abstract has been revised for conciseness and word count updated

3. Really, the study has a clear design of a case control study type however, all the cases of the study group either suspected or probably cases of cutaneous leishmaniasis. All cases diagnosed clinically so in conclusion section, it is better to use “there is evidence” rather than “there is strong evidence”

Agreed. This has been rectified

4. Second: 

a. The pronoun “we” was used several time through the paper, it is better to avoid using the subjective pronouns and it should be dropped from the text.

b. In the line number 93, the abbreviation “CL” was not previously defined and should be written as cutaneous leishmaniasis rather than the short form “CL”.

c. In the line number 81, “intermediate hosted” it is better to be “reservoir” rather than intermediate.

d. In the line number 182, drop the wards “at the” from the sentence

e. In the line number 228, the ward “while” it is better to be dropped.

f. In the line number 262, it is better to add word “Kenia” at the end of the title of table 3.

g. In the line number 290, it is better to add the time of the study “2016” at the end of the title of the table 4.

h. In the line, number 337, it is better to uses “reservoir mammals” rather than “infected wild rodents”

i. In the line number 474, the references 15 should be corrected by 17

j. In the line number 481, the reference 18 should be corrected number 20

Comments a-j have been addressed and updated in the revised manuscript

5. The map should be clearer illustrating all valleys, villages and cities which were mentioned through the paper.

This has been corrected. See new map fig 1

Reviewer #4:

1. General: Abbreviations should be used consistently. CL and cutaneous leishmaniasis are used throughout the manuscript (e.g. in full in line 293). After the first time (line 70?) using the full name the abbreviation should be used.

This has been corrected throughout the document

2. Introduction: Line 93, there are several vectors for CL, which one is meant here?

The species have been clarified and a reference shown

3. Methods: To make what was done easier to understand the paper should consequently discuss the hospital review first (and separately) and the case control study after

The discussion has been re-organized for flow. We first discuss the records review and then present the risk factor analysis. However, some overlapping issues between these two parts of the study have been highlighted under the risk factors for cutaneous leishmaniasis

4. Study population: As a whole, the selection of the study population should be more clearly described. Specifically, the relationship between the line list made based on the hospital records to the case control study. From the 255 only 41 were included as cases. How were the 41 selected? What happened to the rest? Could there have been a bias in selecting these 41? Also for the controls, did all the controls agree to participate in the study? How was the house to house survey used to detect the additional 18 cases carried out? What was the strategy? What were the exact criteria to define cases in the house to house survey.

This has been addressed. 41 probable cases were selected from the outbreak line-list because they could be reached during the study period. Identification of additional cases trough door-to-door case search was a respondent-driven sampling process, whereby study staff working together with village guides would be guided by villagers to identify additional cases. Out of the controls that were approached, 5 were dropped on account of not meeting the criteria for controls. This has been updated in the text and in fig. 2

5. There is mention of suspected, probable or confirmed cases, and that all are included as cases. However, is this correct? How many were actually suspected, probably and confirmed?

Only probable cases were included in the case-control study. The outbreak line-list had a total of 255 suspected and probable cases, out of which 59 probable cases were enrolled in the case control study. No laboratory confirmation was done. See text and updated Fig. 2

6. Were there any patients with missing data? Did this lead to any exclusions?

Patients whose data (e.g. clinical information, physical address, contact information) were missing from the line-list were excluded from enrollment. This has been updated

7. Definition of cases and control: I have a big issue with the definition of cases, as I fear the definition of a CL case is too broad, and therefore the cases contain false positives. Patients with ‘dermatitis’, ‘skin infection’, or ‘skin wound’ now all seem to be recorded as cases, which does not seem correct. This may severely impact the study.

All entries of ‘skin ulcer’, ‘skin wound’, ‘plaque’, ‘dermatitis’, ‘skin infection’ or ‘cutaneous leishmaniasis’ as captured in hospital registers were only included in the outbreak line-list as ‘suspected cases of CL’. For cases that were included in the case-control study, medical officers in the study team had to first ascertain if their skin lesion was a typical CL lesion (ulcer with typical raised edges and depressed center or a skin plaque-a circumscribed palpable or visible lesion made up of multiple coalescent nodules or papules).

8. On line 128 it also states that a probable case was defined as a patient with a typical ulcer (who determines what is a typical ulcer) or plaque, ascertained by a medical officer in the study team during the study period. However, the study period was only 2 weeks, so how can this refer to the part of the study that led to the line list (which to my understanding is based on retrospective file review). In addition, why are only ulcers or plaques mentioned as probable CL, while Sang already stated in 1993 that lesions were rarely ulcerated, and Toroitich also said that many lesions were in fact nodular.

Medical officers in the study team, working together with village locators (community health volunteers and village chiefs) and some of the already recruited cases, identified cases for inclusion in the study during door-to-door visits in a respondent-driven sampling process. These cases were identified on the basis of typical skin lesions (ulcer with raised edges and depressed center or a skin plaque-a circumscribed palpable or visible lesion made up of multiple coalescent nodules or papules) and were updated in the outbreak line-list as ‘probable cases’. The definition of plaques used included skin lesions that were coalescent ‘nodular’ or ‘papular’. This information has been revised in the revised manuscript for clarity.

9. Why was no laboratory confirmation done? Logistical challenges are mentioned, these should be specified. Field work: It states that a lab scientist was there during the field visit, but if no lab confirmation was done, what did this person actually do?

Laboratory confirmation was not done during the outbreak because laboratory supplies needed had not been delivered by the time of the outbreak investigation (sample collection supplies, leishmania skin test supplies and laboratory reagents for tissue staining). Despite having the laboratory scientist in the team, we were not able to conduct any skin tests or collect any samples (skin scrapings/aspirates/biopsies) from the identified cases owing to lack of the necessary supplies. 

10. For the record review, how experienced were the staff in the health facilities in diagnosing CL? How experienced was the medical officer

The study did not gauge the experience of hospital staff (medical doctors, nurses and physician assistants) in diagnosing CL. However, these staff would ordinarily clinically evaluate and capture details of patients with skin lesions in hospital records either as ‘skin ulcer’, ‘skin wound’, ‘plaque’, ‘dermatitis’, ‘skin infection’ or a diagnosis ‘cutaneous leishmaniasis’ in instances when laboratory diagnosis was possible. During review of these hospital records, we abstracted this information to create the outbreak line-list and further updated the line-list by examination of the lesions for patients that could be reached during the 2 weeks of the outbreak response. The two medical officers in the study team had been working in the study area for at least 5 years and had been diagnosing and referring patients with CL for treatment at a specialist clinic located at the referral hospital in Nakuru town.

11. Matching: Why was matching by neighborhood done? This does not make sense when factors related to the outdoor environment are one of the main interests of the study. By matching by neighbourhood, neighbourhood itself (micro-environment) cannot be studied properly as a risk factor in that way. �I think the third level of factors related to outdoor environment cannot be properly studied if neighbors are chosen as controls (unless the houses are very far apart, which should then be adequately explained). Because no direct neighbors of ‘CL cases’ could be taken as controls, how can having an immediate neighbor with CL be studied as a risk factor? Why matching when multivariable regression is done in analysis to deal with confounding anyway?

The study area is relatively sparsely populated (Population density of 148 persons per sq. Km), with most households located beyond 150-500 meters apart, sometimes separated by thickets, valleys, or large farms (See description of the study site in the revised manuscript). To get eligible controls, the study team would skip between 2 and 5 households in a chosen direction before approaching a potential control for recruitment. Therefore, many times the controls would land in the next village, far from the location of the case and would not necessarily share the same micro-environment. 

Matching by age-group and neighborhood at design level of this study does not necessarily control for all confounding, hence the need for us to still conduct a multivariate regression to control for any residual confounding. On analysis of matched case-control studies, Neil Pearce recommends that matching alone does not eliminate confounding, and may still require controlling of the matching factors in the analysis (Neil Pearce- Analysis of matched case-control studies, BMJ 2016 available from https://www.bmj.com/content/bmj/352/bmj.i969.full.pdf )

12. Sample size: The manuscript states that OpenEpi was used to calculate the sample size. However, OpenEpi does not seem to have an option for calculating sample size for matched case control studies. When I redo the sample size calculation in open epi using unmatched case control I get 180, 167 or 189, not 174. It should be clearly indicated how they reached 174. After trying to repeat the sample size calculation it seems that a sample size calculation for matched case control assuming correlation is 0.15 was used. However, this is not explained clearly, and if 0.15 correlation was used, this should be explained. Sample size uses assumption that mosquito net use was present in 31% of controls. In fact this was <20%. I think this may lead to your study being underpowered, as when I retrospectively calculate the sample size using the 18% proportion controls using a mosquito net, the sample size is 306, 279 or 317, which is almost double of the current sample size.

The numbers of eligible cases present in the study area during the study period (58 cases) largely determined the sample size. A minimum sample size of 174 (2 controls per case) was used, giving the study had at least 80% power at the 5% significance level and able to detect an odds ratio (OR) of 0.3 for an exposure present in 31% of controls. We used OpenEpi to calculate sample size for unmatched case-control studies. This information has been updated in the revised manuscript 

13. Ethics: This study was not reviewed by any ethical review body. The reason given is that the study was conducted as part of an MOH led public health response to an acute event. However, to my understanding even research carried out in emergency response (e.g. in Ebola research) requires (expedited) ethical review. The researchers should elaborate more on the reason and regulations as to why the study was not reviewed by any ethical board. Just because permission was given by the health authorities does not automatically mean the study is carried out according to the ethical principles of medical research. For instance, regarding the consent procedures (see below). The manuscript states that oral consent was obtained. However, there is no mention of any proof of this consent (e.g. recording of oral consent, or ideally written consent). It is also not stated clearly why for this study oral (non-recorded) consent suffices. I have my doubts about this, and therefore the reasons for these matters should be explicitly explained. In addition, the informed consent should be explained in more detail (especially since it was not approved by any IRB), was the consent procedure organized in an appropriate and complete manner? Did the patient receive a participant information sheet describing the research and giving the contact information of the PI?

This study was approved by Ministry of Health (MOH) in Kenya and was conducted as part of public health response to an acute event and as such was not reviewed by an ethical review body. Oral consent was obtained from the case-control study subjects and documented in the study questionnaires as indicated in the revised manuscript. Study information was provided in form of written leaflets given to all study participants and printed brochures displayed at strategic locations in health facilities. The records review and active community case finding were conducted as part of routine surveillance and response activities by the MOH. In instances of acute events (such as outbreaks), the practice is that outbreak investigation, response and regulatory oversight is offered by the government Ministry of Health. However, approvals from various levels of administration (e.g. county and subcounty levels) are still mandatory and were sought by the response team. All the data collected was anonymized by dropping all personal identifiers (e.g. name, inpatient/outpatient number, phone contact) before analysis.

14. Measures were taken to assure confidentiality. Please specify which methods.

Measures taken to assure confidentiality of the information provided during these interviews included conducting interviews in private place convenient for respondents, secure storage of paper questionnaires in lockable cabinets accessible only with authorization of the PI and password protection of electronic databases. These have been updated in the revised manuscript.

15. The manuscript states that no personal identifying information was collected or analysed. However, in line 133 it states that name, sex, age, date seen at the facility, residence, signs and symptoms, diagnosis and treatment given were all among collected data. This would definitely qualify as personal identifying information.

Patient data captured in health facility outbreak line-list but was de-identified by dropping all the personal identifying information (patient name, IP/OP number, phone contact) before analysis.

16. Why use the 3 steps in multivariate? Why not skip the analysis per level?

Grouping of the risk factors into three categories was necessary since a number of the factors being explored were potentially inter-related and their parameters would likely vary at more than one level (i.e. multilevel). For example, individual factors would affect a person’s exposure to and vulnerability to sand-fly bites, indoor factors would affect the abundance of sandflies in the house and/or vulnerability to bites while outdoor factors would affect abundance and likelihood of bites when the individual is outside the residence. An individual (with his factors) would ordinarily be nested within an indoor or outdoor environment that has a mix of related factors, some of which are mutually exclusive in this study e.g. examining the relationship between age (individual factor), spending time within the house (indoor factor and is likely associated with individual’s age) and spending time outside the residence (outdoor factor that is related to age and is mutually exclusive of spending most time indoors) gives an example of how mutual exclusivity of the factors would affect the levels of analysis.

17. Results: 

a. It is mentioned that one death was suspected due to CL. This is strange as CL is normally not deadly, and therefore this should be elaborated on.

The death was reported in a 67-year old woman who had undergone weeks of treatment for CL

b. Line 203: IQR should give the 25th and 75th percentile, and not just one value)

c. Line 218-219 should be left out, as matching means that the age is very similar

d. Line 220: states that distribution of other social and demographic variables studied among cases and controls was comparable. However, occupation was significantly different between the two groups.

e. Line 227: mean duration of illness was two years .

f. Line 228: it says all cases had typical ulcers, but it does not make sense to describe a feature that was used to select the population. If typical ulcers were the way the population was selected, of course all patients will have typical ulcers.

b-f have been addressed in the revised manuscript

g. The number of lesions should be mentioned. If the big majority of cases had both a typical ulcer but also a papule or nodule, does this mean they had multiple lesions with different presentations or one lesion with both an ulcerative and indurated presentation?

The majority of cases had multiple lesions with different skin presentations, mainly ulcerative and nodular lesions. This statement has been revised

h. Line 232: why mention nasal stuffiness? This seems random, unless you want to link it MCL.

Nasal stuffiness was observed in 2 patients that had severely ulcerative lesions affecting the peri-nasal area.

i. Line 238-244. If patients were close neighbors how can there be a difference in sighting rock hyraxes around the house or in living close to the forest? Could this be purely due to bias?

Rock hyraxes are territorial animals, live is specific rocky habitat and do not forage far (>50 meters) from their residence. In the study area, the population density is low and houses are located up to 200 meters from each other. I therefore believe that sighting rock hyraxes around the residence is a valid question for this study.

j. 252: here 2 decimals are given, throughout the manuscript it should be consistent, either 1 or two decimals for percentages and CI.

This has been addressed

k. Line 257: certain activities were not analysed because they could potentially have a close link with forest visits. However, they should be analyzed in multivariable which can check whether they are independent risk factors.

I would still be hesitant to analyze each of these factors separately in the multivariate model since the underlying denominator seems to be visiting the forest (which was identified as an independent risk factor for CL in the regression model). It would be complex to interpret the aORs for each of these activities separately, when put in the same model with ‘forest visits’

l. Are the factors displayed in table 3 the only ones that were assessed? Or did a selection take place and only certain ones are displayed? This should be described.

We reported on all the factors that were assessed except for the activities such as charcoal burning, hunting, herding, stone masonry and mining that were included among the occupations involving forest visits. It has been explained that these factors had significantly large ORs but were not reported here due to possible close link between each of these occupations with forest visits (see revised manuscript with explanation)

m. Table 2 and 3 can be merged into one, as there is significant overlap.

The two tables communicate different information from our analysis. Table 2 shows the overall demographic characteristics of the study respondents while table 3 presents the results of the bivariate analysis for the factors that were assessed. The two tables were presented separately for simplicity and clarity of the two messages

n. Line 264: should bivariate analysis not be multivariate here?

Corrected

o. The results from the risk factor analysis can be shortened. The essence should be which factors were risk factors in the bivariate, and how many of them were independent risk factors after multivariate analysis?

This section has been revised

18. Discussion: 

a. I feel the burden of CL has not been sufficiently highlighted. At least, more background should be given that can help quantify the burden in relation to the study area.

This has been addressed by addition of more background information about the study area and revision of first paragraph of the discussion.

b. Limitations regarding recall bias and the case definition should be mentioned.

c. Line 310: this seems to be over interpretation of the findings, this study does not allow saying anything about indoor or outdoor transmission being observed.

b and c have been revised

d. The CL outbreak of 2016 is never mentioned again. Was this actually an outbreak? In line 355 it mentioned there may be accelerated transmission. A reference from 1993 is given. This indicates that CL is not a new thing here and therefore points in the other direction in fact.

Indeed, CL has been known to be present in this geographical area since the early 1990’s as seen from the work done by Rosemary Sang (1993) and Toroitich (1995). In our investigation, we found a continuous transmission pattern of cases between 2010-2016 without any obvious peaks that would characterize an outbreak. We therefore concluded that this was a case of accelerated transmission owing to existence of multiple factors of spread (indoor and outdoor) in the area.

e. Line 369: it is not clear whether the statement of low levels of education is about the cases in the case control study, the hospital records or in the population comprising both cases and controls. In addition the 90.8% that is mentioned to have less than primary education can not be found in any of the tables, which mention 94.8% of cases having primary education or less (not less than primary education).

This has been reviewed. The correct percentage for primary level of education and below is 90.8%

f. Line 375: patients above 35 are mentioned being ‘older individuals.’ This statement is related to immunosuppression of old age. This is not appropriate, as only 10% of patients were above 60 in fact.

This has been corrected. ‘Older individuals’ has been dropped from this statement

g. Line 386: states the most common presentation of cases was a skin ulcer, if this was the way the cases were selected do not use it to describe the presenting feature of CL.

h. The limitations of the case control study should be elaborated on, as there are quite a few (see comments from methods section).

g and h have been revised

19. Conclusions:

a. 404: no real grounds for this statement. What is large?

b. 407: study does not provide real evidence on transmission, just on risk factors

c. 414: what is meant by ‘there is need to institute surveillance for CL’?

I have revised this section in line with the revisions in the rest of the manuscript

20. References: Please correct the mistakes here, I found a few

a. -ref 9: should be Sang D, Ashford RW, Njeru WK etc.

b. -ref 10: authors name is Toroitich AK (not Itor Toroitich)

c. -ref 11: third author’s first rather than last name is written in full

These have been addressed

---

## [Decision Letter · Decision Letter 1]

2 Dec 2019

<h4>**PONE-D-19-19202R1**

**Burden and risk factors of Cutaneous Leishmaniasis in a peri-urban settlement in Kenya, 2016**

**PLOS ONE**

**Dear Dr Ngere,**

**Thank you very much for submitting your manuscript "Cases of Cutaneous Leishmaniasis in a peri-urban settlement in Kenya, 2016" (#PONE-D-19-19202R1) for review by PLOS ONE. As with all papers submitted to the journal, your manuscript was fully evaluated by academic editor (myself) and by independent peer reviewers. The reviewers appreciated the attention to an important health topic, but they raised substantial concerns about the paper that must be addressed before this manuscript can be accurately assessed for meeting the PLOS ONE criteria. Therefore, if you feel these issues can be adequately addressed, we invite you to submit a revised version of the manuscript that addresses the points raised during the review process. We can’t, of course, promise publication at that time.**

**We would appreciate receiving your revised manuscript by Jan 16 2020 11:59PM. When you are ready to submit your revision, log on to**
https://www.editorialmanager.com/pone/
**and select the 'Submissions Needing Revision' folder to locate your manuscript file.**

****

**To enhance the reproducibility of your results, we recommend that if applicable you deposit your laboratory protocols in protocols.io, where a protocol can be assigned its own identifier (DOI) such that it can be cited independently in the future. For instructions see:**
http://journals.plos.org/plosone/s/submission-guidelines#loc-laboratory-protocols

**Please include the following items when submitting your revised manuscript:**</h4>

<h4>**A rebuttal letter that responds to each point raised by the academic editor and reviewer(s). This letter should be uploaded as separate file and labeled 'Response to Reviewers'.**</h4><h4>**A marked-up copy of your manuscript that highlights changes made to the original version. This file should be uploaded as separate file and labeled 'Revised Manuscript with Track Changes'.**</h4><h4>**An unmarked version of your revised paper without tracked changes. This file should be uploaded as separate file and labeled 'Manuscript'.**</h4>

<h4>**Please note while forming your response, if your article is accepted, you may have the opportunity to make the peer review history publicly available. The record will include editor decision letters (with reviews) and your responses to reviewer comments. If eligible, we will contact you to opt in or out.**</h4>

<h4>**We look forward to receiving your revised manuscript.**

**Kind regards,**

**Abdallah M. Samy, PhD**

**Academic Editor**

**PLOS ONE**

**Reviewers' comments:**

**Reviewer's Responses to Questions**

**Comments to the Author**

**1. If the authors have adequately addressed your comments raised in a previous round of review and you feel that this manuscript is now acceptable for publication, you may indicate that here to bypass the “Comments to the Author” section, enter your conflict of interest statement in the “Confidential to Editor” section, and submit your "Accept" recommendation.**

**Reviewer #1: (No Response)**

**Reviewer #3: All comments have been addressed**

**Reviewer #4: (No Response)**</h4>

<h4>**2. Is the manuscript technically sound, and do the data support the conclusions?**

**The manuscript must describe a technically sound piece of scientific research with data that supports the conclusions. Experiments must have been conducted rigorously, with appropriate controls, replication, and sample sizes. The conclusions must be drawn appropriately based on the data presented. **

**Reviewer #1: No**

**Reviewer #3: Yes**

**Reviewer #4: Yes**</h4>

<h4>**3. Has the statistical analysis been performed appropriately and rigorously? **

**Reviewer #1: No**

**Reviewer #3: Yes**

**Reviewer #4: I Don't Know**</h4>

<h4>**4. Have the authors made all data underlying the findings in their manuscript fully available?**

**The**
PLOS Data policy
**requires authors to make all data underlying the findings described in their manuscript fully available without restriction, with rare exception (please refer to the Data Availability Statement in the manuscript PDF file). The data should be provided as part of the manuscript or its supporting information, or deposited to a public repository. For example, in addition to summary statistics, the data points behind means, medians and variance measures should be available. If there are restrictions on publicly sharing data—e.g. participant privacy or use of data from a third party—those must be specified.**

**Reviewer #1: Yes**

**Reviewer #3: No**

**Reviewer #4: Yes**</h4>

<h4>**5. Is the manuscript presented in an intelligible fashion and written in standard English?**

**PLOS ONE does not copyedit accepted manuscripts, so the language in submitted articles must be clear, correct, and unambiguous. Any typographical or grammatical errors should be corrected at revision, so please note any specific errors here.**

**Reviewer #1: No**

**Reviewer #3: Yes**

**Reviewer #4: Yes**</h4>

<h4>**6. Review Comments to the Author**

**Please use the space provided to explain your answers to the questions above. You may also include additional comments for the author, including concerns about dual publication, research ethics, or publication ethics. (Please upload your review as an attachment if it exceeds 20,000 characters)**

**Reviewer #1: Abstract**

**“Of the 255 suspected cases of cutaneous leishmaniasis identified, females constituted 56% 28 (142/255) and the median age was 7 years (IQR 7-21). Cases were clustered in two locations: Gitare (28.6%, 73/255) and Kambi-Turkana (14%, 36/255) and a continuous transmission 30 pattern throughout the period was depicted”**

**The authors mentioned 255 cases. Then, they mention 73 case in Gitare and 36 case in Kambi giving a total of 109 case! This is really confusing!**

**Methodology**

**Sample recruiting is very confusing and I was not satisfied with the explanation given to my previous concerns. In line number 141, the authors mentioned 172 cases. This number (172) was never mentioned again!**

**They recruited 41 cases from previous medical records assuming that those cases are CL. The medical record did not mention the diagnosis clearly and authors inferred from the description of lesions that those patients had CL. If the medical personnel who treated the patients when they visited the hospital could make the diagnosis, how authors could do so!**

**Recruiting samples is very weak and hence the conclusions cannot be reliable, unfortunately.**

**Reviewer #3: Thank you for the revised revision, however, other comment need to be addressed**

**In the references section.**

**a) In the line 477, the reference number should be corrected to be 10 rather than 11**

**b) In the line 501, the reference number should be corrected to be 19 rather than 15.**

**c) In the line 508, the reference number should be corrected to be 22 rather than 18**

**Reviewer #4: Dear author,**

**The manuscript has improved significantly and the link between the hospital based study and case control study is clearer now. The limitations have also been made more clear.**

**There are still a few things I would like to point out**

**-The CL abbreviation is still not done correctly (e.g. line 64, line 70, line 216)**

**-Be consistent in formatting e.g. rift valley (92), Rift Valley (85), Sub-county (98) or sub county (100)**

**-Typo's : line 98 (the study 2x), line 443-444 ("would be in theory be effective but would not be practical, (102) larva should be 'lava'**

**-The statement on line 68-69 is from WHO I think (not from the review so change reference) and is also incorrect, an estimated 2 million new infections with leishmania (including 0.5 mil for VL) occur yearly, of which 1.5 million are for CL**

**-The sample size is still not clear to me based on the new version of the manuscript and the reply to the comments. It seems all cases that could be found were included, and then a power calculation was done afterwards, rather then a sample size calculation beforehand, which would have guided when to stop recruiting patients. If this is the case I would phrase it more like below for clarity**

**"Based on the 58 patients and 116 controls (2 controls per case) , the study had 80% power to detect an odds ratio of 0.3 (using the 5% significance level for mosquito nets which was estimated to be present in 31% of controls."**</h4>

<h4>**7. PLOS authors have the option to publish the peer review history of their article (what does this mean?). If published, this will include your full peer review and any attached files.**

****

****

**Reviewer #1: No**

**Reviewer #3: Yes: Mohammed Musid Saad Alkulaibi**

**Reviewer #4: No**

****

**While revising your submission, please upload your figure files to the Preflight Analysis and Conversion Engine (PACE) digital diagnostic tool, https://pacev2.apexcovantage.com/. PACE helps ensure that figures meet PLOS requirements. To use PACE, you must first register as a user. Registration is free. Then, login and navigate to the UPLOAD tab, where you will find detailed instructions on how to use the tool. If you encounter any issues or have any questions when using PACE, please email us at figures@plos.org. Please note that Supporting Information files do not need this step.**</h4>

---

## [Author Response · Author response to Decision Letter 1]

3 Dec 2019

REVIEWER #1.

1. Abstract

“Of the 255 suspected cases of cutaneous leishmaniasis identified, females constituted 56% 28 (142/255) and the median age was 7 years (IQR 7-21). Cases were clustered in two locations: Gitare (28.6%, 73/255) and Kambi-Turkana (14%, 36/255) and a continuous transmission 30 pattern throughout the period was depicted”

The authors mentioned 255 cases. Then, they mention 73 case in Gitare and 36 case in Kambi giving a total of 109 case! This is really confusing!

Response: This statement has been revised to read “Cases occurred in clusters with up to 43% of cases originating from Gitare (73/255) and Kambi-Turkana (36/255) villages and a continuous transmission pattern throughout the period was depicted”

2. Methodology

Sample recruiting is very confusing and I was not satisfied with the explanation given to my previous concerns. In line number 141, the authors mentioned 172 cases. This number (172) was never mentioned again!

They recruited 41 cases from previous medical records assuming that those cases are CL. The medical record did not mention the diagnosis clearly and authors inferred from the description of lesions that those patients had CL. If the medical personnel who treated the patients when they visited the hospital could make the diagnosis, how authors could do so! Recruiting samples is very weak and hence the conclusions cannot be reliable, unfortunately.

Response: The section outlining recruitment of cases and controls and sample size estimation has been re-organized for clarity and flow. The sample size was determined by the number of eligible cases present in the study area during the study period. A sample size of 58 patients and 116 controls (2 controls per case) was therefore adopted to give the study at least 80% power at the 5% significance level and able to detect an odds ratio (OR) of 0.3 for an exposure present in 31% of controls. All the 41 cases that were selected from the outbreak line-list were first ascertained to be probable CL on the basis of typical skin lesions (a skin ulcer or a plaque) by experienced medical officers who were part of the study team.

REVIEWER #3.

1. Thank you for the revised revision, however, other comment need to be addressed

In the references section.

a. In the line 477, the reference number should be corrected to be 10 rather than 11

b. In the line 501, the reference number should be corrected to be 19 rather than 15

c. In the line 508, the reference number should be corrected to be 22 rather than 18

Response: The numbering has been corrected in the reference section of the revised manuscript

REVIEWER #3.

The manuscript has improved significantly and the link between the hospital based study and case control study is clearer now. The limitations have also been made more clear. There are still a few things I would like to point out

1. The CL abbreviation is still not done correctly (e.g. line 64, line 70, line 216)

2. Be consistent in formatting e.g. rift valley (92), Rift Valley (85), Sub-county (98) or sub county (100)

3. Typo's : line 98 (the study 2x), line 443-444 ("would be in theory be effective but would not be practical, (102) larva should be 'lava'

Response: These typos have been corrected

4. The statement on line 68-69 is from WHO I think (not from the review so change reference) and is also incorrect, an estimated 2 million new infections with leishmania (including 0.5 mil for VL) occur yearly, of which 1.5 million are for CL

Response: This reference has been corrected

5. The sample size is still not clear to me based on the new version of the manuscript and the reply to the comments. It seems all cases that could be found were included, and then a power calculation was done afterwards, rather then a sample size calculation beforehand, which would have guided when to stop recruiting patients. If this is the case I would phrase it more like below for clarity

"Based on the 58 patients and 116 controls (2 controls per case) , the study had 80% power to detect an odds ratio of 0.3 (using the 5% significance level for mosquito nets which was estimated to be present in 31% of controls."

Response: This section has been re-written for clarity and flow. The sample size was determined by the numbers of eligible cases present in the study area during the study period. A sample size of 58 patients and 116 controls (2 controls per case) was adopted to give the study at least 80% power at the 5% significance level and able to detect an odds ratio (OR) of 0.3 for an exposure present in 31% of controls. The exposure chosen was use of mosquito nets.

All other comments and typographical errors have been corrected

---

## [Editor Report · Decision Letter 2]

5 Dec 2019

PONE-D-19-19202R2

Burden and risk factors of Cutaneous Leishmaniasis in a peri-urban settlement in Kenya, 2016

PLOS ONE

Dear Dr Ngere,

Thank you for submitting your manuscript to PLOS ONE. After careful consideration, we feel that it has merit but does not fully meet PLOS ONE’s publication criteria as it currently stands. Therefore, we invite you to submit a revised version of the manuscript that addresses the points raised during the review process.

We would appreciate receiving your revised manuscript by Jan 19 2020 11:59PM. To enhance the reproducibility of your results, we recommend that if applicable you deposit your laboratory protocols in protocols.io, where a protocol can be assigned its own identifier (DOI) such that it can be cited independently in the future. For instructions see: http://journals.plos.org/plosone/s/submission-guidelines#loc-laboratory-protocols

We look forward to receiving your revised manuscript.

Kind regards,

Abdallah M. Samy, PhD

Academic Editor

PLOS ONE

**Comments:**

- The authors should provide the figures in 300 dpi as per Journal regulations; all figures looks unsuitable as it currently stands. 

- Table 3: :"Net use by the individual", do you mean the bednet? 

- Please carefully revise the language of your manuscript in all sections and tables.  

---

## [Author Response · Author response to Decision Letter 2]

17 Dec 2019

- The authors should provide the figures in 300 dpi as per Journal regulations; all figures looks unsuitable as it currently stands. 

This has been addressed. The figures have been edited to the right format and uploaded via the PACE digital diagnostic tool for compliance to journal requirements

- Table 3: :"Net use by the individual", do you mean the bednet? 

This table has been edited to read ‘individual using bed net’

- Please carefully revise the language of your manuscript in all sections and tables. Other variable names in the table have been revised accordingly

We have revised the language and flow of the entire manuscript, including the tables.

All other comments and typographical errors have also been corrected

---

## [Editor Report · Decision Letter 3]

27 Dec 2019

Burden and risk factors of cutaneous leishmaniasis in a peri-urban settlement in Kenya, 2016

PONE-D-19-19202R3

Dear Dr. Ngere,

We are pleased to inform you that your manuscript has been judged scientifically suitable for publication and will be formally accepted for publication once it complies with all outstanding technical requirements.

With kind regards,

Abdallah M. Samy, PhD

Academic Editor

PLOS ONE

---

## [Editor Report · Acceptance letter]

31 Dec 2019

PONE-D-19-19202R3 

Burden and risk factors of cutaneous leishmaniasis in a peri-urban settlement in Kenya, 2016 

Dear Dr. Ngere:

I am pleased to inform you that your manuscript has been deemed suitable for publication in PLOS ONE. Congratulations! Your manuscript is now with our production department. 

With kind regards,

on behalf of

Dr. Abdallah M. Samy 

Academic Editor

PLOS ONE